# Effects of Barley Starch Level in Diet on Fermentation and Microflora in Rumen of Hu Sheep

**DOI:** 10.3390/ani12151941

**Published:** 2022-07-30

**Authors:** Zhian Zhang, Fei Li, Xiaowen Ma, Fadi Li, Zongli Wang

**Affiliations:** State Key Laboratory of Grassland Agro-Ecosystems, Key Laboratory of Grassland Livestock Industry Innovation, Ministry of Agriculture and Rural Affairs, Engineering Research Center of Grassland Industry, Ministry of Education, College of Pastoral Agriculture Science and Technology, Lanzhou University, Lanzhou 730020, China; zhangzha20@lzu.edu.cn (Z.Z.); maxw18@lzu.edu.cn (X.M.); lifd@lzu.edu.cn (F.L.); wangzongli@sina.com (Z.W.)

**Keywords:** starch source, rumen fermentation, microflora, Hu sheep

## Abstract

**Simple Summary:**

Increasing the starch level in the diet of ruminants can promote rumen fermentation to a certain extent and improve production performance. However, the degradation of rapidly fermentable carbohydrates in the rumen can lead to the accumulation of large amounts of organic acids (mainly volatile fatty acids and lactic acid), increasing the risk of subacute ruminal acidosis (SARA) in high-yielding ruminants. By changing the fermentation sites of rapidly fermentable carbohydrates, the rumen environment and feed efficiency could be effectively improved. The object of this study was to explore the effects of replacing corn starch with barley starch in the diet of Hu sheep on the fermentation and microflora in the rumen. Our results showed that, starch in the diet cannot be derived entirely from barley, and it is feasible to replace 33% of the corn with barley in the diet for Hu sheep.

**Abstract:**

This study aimed to explore the effects of different levels of barley starch instead of corn starch on the rumen fermentation and microflora when feeding a corn-based diet to Hu sheep. Thirty-two male Hu sheep equipped with permanent rumen fistulas were selected and fed in individual metabolic cages. All sheep were randomly divided into four groups (eight sheep in each group) and fed with four diets containing a similar starch content, but from different starch sources, including 100% of starch derived from corn (CS), 33% of starch derived from barley + 67% of starch derived from corn (33 BS), 67% of starch derived from barley + 33% of starch derived from corn (67 BS) and 100% of starch derived from barley (100 BS). The experimental period included a 14 d adaptation period and a 2 d continuous data collection period. The results showed that the molar proportions of acetate, isobutyrate, butyrate and isovalerate and the ratio of acetate to propionate in the 67 BS and 100 BS groups decreased compared with the CS and 33 BS groups (*p* < 0.001), while the molar proportions of propionate and valerate increased (*p* < 0.001). The combination of 33% barley starch and 67% corn starch in the diet improved the production of TVFAs (*p* = 0.007). The OTUs and Shannon indexes of the CS and 33 BS groups were higher than the 67 BS and 100 BS groups (*p* < 0.001), and the Chao1 and Ace indexes were higher than the 100 BS group (*p* < 0.05). In addition, the 33 BS group had increased the relative abundances of *Bacteroidetes*, *Prevotella* and *Ruminococcus* and the abundances of *Fibrobacter succinogenes*, *Ruminococcus flavefaciens*, *Streptococcus bovis*, *Selenomonas ruminantium* and *Prevotella brevis* relative to the CS group (*p* < 0.05). These results indicate that the substitution of 33% of the CS with BS did not change the rumen fermentation pattern relative to the CS group, and increased the richness and diversity of the rumen microbes in Hu sheep compared with other two starch substitute groups.

## 1. Introduction

In the current intensive ruminant farms, increasing the proportion of rapidly fermentable carbohydrates in the diet has become a common strategy for improving the performance of ruminants. However, the rapid ruminal digestion of starch in the rumen also increases the risk of subacute rumen acidosis (SARA) [1], especially for high-yielding cows, goats and beef cattle [2,3,4]. The starch content of corn grain is 62.6% (dry matter), which is currently one main source of starch in ruminant diets. However, barley has a higher amylopectin content than corn, which can be fermented and digested in the rumen to a greater extent than corn and could be a potential replacement for corn as a source of energy for ruminants [5,6]. Combining barley starch (BS) with corn starch (CS) could improve the efficiency of starch digestion by balancing starch fermentation in the rumen or small intestine [3,7,8].

Several studies have shown that high-grain diets can cause abnormal rumen fermentation in ruminants, producing a large amount of short-chain volatile fatty acids or lactic acid in the rumen, thereby reducing rumen pH and altering the abundance of cellulolytic bacteria [6,9]. Mccann et al. showed that the rapid degradation of barley grain in the rumen resulted in an accumulation of volatile fatty acids (VFAs), which reduced the rumen’s pH and increased the risk of subacute ruminal acidosis (SARA) [5]. Li et al. found that the molar proportion of acetate decreased, whereas that of propionate increased with an increased replacement proportion of corn with wheat [10]. In addition, most studies usually involve one source of non-structural carbohydrates being completely replaced by another source of carbohydrates, but research data about the effect of different ratios of two non-structural carbohydrates in the diet of sheep on rumen fermentation and microflora are relatively limited [11,12]. Our previous study has shown that replacing CS with BS in the diet affected the performance, nutrient digestion and rumen fermentation of Hu sheep [8]; one main reason for this might be due to changes in the rumen microflora. However, insufficient attention was paid to the effects on rumen microflora in the previous study; a few bacteria were determined, which could not reflect an alteration of the rumen’s microbial structure. In addition, Ma et al. also found that, within a day of equipping sheep with fistulas, the mean pH decreased with increasing dietary proportions of BS, which would change microbial structure in the rumen [8]; unfortunately, there was no relevant information to confirm this. Further research is needed to clarify the effects of replacing corn starch with barley starch on the rumen’s microbial diversity and richness, and on the bacteria that dominate at the phylum or genus level.

Therefore, we hypothesized that increasing the ratio of barley in the diet would promote rumen fermentation, increasing the TVFA concentration and the molar concentration of propionate, reducing the rumen’s pH, and altering the composition and structure of the rumen’s microbial community. The object of this study was to explore the effects of replacing corn starch with barley starch in the diet of Hu sheep on the dynamic fermentation and microflora in the rumen, which will be beneficial for providing some new insights into the efficient utilization of feed and the rumen health of sheep.

## 2. Materials and Methods

### 2.1. Animals, Diets, and Experimental Design

Thirty-two male Hu sheep, surgically fitted with ruminal fistulas, were selected for this experiment. All of the sheep were raised in individual metabolic cages (0.75 m × 1.5 m × 1.0 m), fed twice per day at 0800 and 1800 h in amounts that allowed 5% refusals, and had free access to water. According to dietary treatments, the sheep were randomly divided into four treatment groups, including a CS group (all starch in the diet derived from maize), a 33 BS group (33% of barley starch + 67% of corn starch), a 67 BS group (67% of barley starch + 33% of corn starch), and a 100 BS group (all starch in the diet derived from barley). The diets in the different treatments were formulated to contain similar amounts of starch, but the proportion of starch supplied by corn and barley varied. Ingredients and the chemical composition of the pellet-TMR diets are shown in Table 1. The experimental period included a 14 d adaptation period and a 2 d continuous sampling and data-collecting period. The amounts of feed offered in dry matter form (5% of BW) and the orts were recorded daily for the calculation of dry matter intake (DMI).

### 2.2. Sample Collection

After the adaptation period, samples of rumen fluid at 0 h, 2 h, 4 h, 6 h, 8 h and 10 h after the morning feeding were collected. The rumen digesta was removed from the fistula and filtered through 4 layers of cheesecloth with a mesh size of 250 μm, and the pH of the filtrate was immediately determined using a portable pH meter (PHB-4, Shanghai Hongyi Instrument Limited, Shanghai, China). Eight milliliters of filtered rumen fluid in a screw-top centrifuge tube was mixed with 2 mL of 250 g/L of metaphosphoric acid and stored at −20 °C for the determination of volatile fatty acids in the rumen of Hu sheep. Another 10 mL of the rumen fluid sample was immediately stored at −80 °C to minimize any possible microbial activities, for use in later DNA extractions.

### 2.3. Fermentation and Feed Composition Analysis

The short-chain fatty acid concentration was determined by using a gas chromatograph (Thermo Fisher Scientific, Milano, Italy) with a DB-FFAP capillary column (DB-FFAP, 30 m × 0.32 mm × 0.25 µm, Agilent Technologies Co., Ltd., Santa Clara, CA, USA) using the method described by Li et al. [13]. The feed samples were dried at 65 °C in a forced-air oven for 72 h and air-equilibrated for 12 h, then ground using a 1 mm screen. The feed samples were analyzed for DM (dried at 135 °C for 3 h in a forced-air oven); CP (AOAC 2000, Kjeldahl method 988.05); starch (using a commercial assay kit; Jiancheng Bioengineering Institute, Nanjing, China); and NDF and ADF using heat-stable alpha-amylase and sodium sulfite, following the methods of Van Soest et al. [14].

### 2.4. Extraction of Bacterial DNA and Illumina MiSeq Sequencing

Six samples (2 h after the morning feeding) were selected randomly from each group for the genomic DNA extraction of rumen fluid using the methods of Li et al. [15]. A Nanodrop 1000 Nucleic Acid Quantifier (Thermo Fisher Scientific, Wilmington, DE, USA) was used to detect DNA concentration and purity. Quantitative analysis of the microorganisms in the rumen was performed with a Bio-Rad CFX96 Real-Time System (Bio-Rad Laboratories, Hercules, CA, USA), according to the methods described by Li et al. [10] and Zhang et al. [16]. The primer sets used for bacterial PCR amplification are presented in Table 2 [16]. The results for the counting of each species were expressed as log10 copy number of 16S rRNA gene copies per mL of rumen fluid.

Illumina MiSeq sequencing was conducted at the Beijing Genomics Institute (Shenzhen, China). The bacterial universal primers used to amplify V3–V4 were 5′-ACTCCTACGGGAGGCAGCA-3′ (forward primer) and 5′-GGACTACHVGGGTWTCTAAT-3′ (reverse primer), which covered the V3 to V4 regions of the bacterial 16S rRNA genes. The steps were as follows: take 30 ng of qualified genomic DNA samples and the corresponding fusion primers to configure the PCR reaction system; set the PCR reaction parameters for PCR amplification; use Agencourt AMPure XP magnetic beads to purify the PCR amplification products and dissolve them in elution buffer, and label them; complete the library construction. The Agilent 2100 Bioanalyzer was used to detect the fragment range and concentration of the library. Qualified libraries were selected for sequencing on the HiSeq platform according to the size of the inserted fragments. Data were filtered, and the remaining high-quality clean data were used for later analysis (minimum overlap of 15 bp, maximum mismatch rate set to 0.1). The FLASH (Fast Length Adjustment of SHort reads, v1.2.11) software was used to assemble the paired reads obtained from paired-end sequencing into a sequence by using the overlap relationship. The overlap relationship between the reads was spliced into tags; tags were clustered into OTUs and compared with the database and species annotations. USEARCH (v7.0.1090) was used to cluster at 97% similarity to obtain the representative sequence of OTUs. UCHIME (v4.2.40) was utilized to remove the chimera generated by PCR amplification from the OTU-representative sequence, and the usearch_global method was used to align all the tags back to the OTU-representative sequence to obtain the OTU abundance statistics table for each sample. Then, these were aligned against entries in the Silva alignment database (release 128). Based on the OTU and annotation results, a statistical analysis of gene abundance was conducted at the corresponding taxonomical level (phylum and genus). To compare the dissimilarities in bacterial communities among the four groups, a principal coordinate analysis (PCoA) based on weighted UniFrac distances at the OTU level was performed. All raw sequences were deposited in the NCBI Sequence Read Archive (SRA) database and can be accessed via accession number: PRJNA859264.

### 2.5. Statistical Analysis

The data were preliminarily sorted through Excel 2010, and SPSS 21.0 software (SPSS, Chicago, IL, USA) was used to analyze the effects of sampling time, barley ratio in the diet and the interaction between them on the rumen fermentation parameters by using the general linear model. The quantification of rumen microbes and the difference analyses of α-diversity and β-diversity between each group were performed by one-way ANOVA (analysis of variance), and the results were expressed as the average value. Significance was declared at *p* < 0.05, and trends were declared at 0.05 < *p* < 0.10.

## 3. Results

### 3.1. Daily Total DMI, Ruminal pH, and VFAs

There were no effects of the interaction between sampling time and treatment on the rumen pH, total volatile fatty acid (TVFA) content or the molar proportions of other VFAs (*p* > 0.05, Table 3). The TVFA content tended to be the lowest (*p* = 0.071), while rumen pH and the molar proportions of acetate, isobutyrate and isovalerate were the highest (*p* < 0.001), prior to the morning feeding, and they were affected by sampling time. The content of TVFA in the 33 BS group was higher than in the CS and 67 BS groups (*p* = 0.007). The molar proportion of acetate and ratio of acetate to propionate in the CS and 33 BS groups increased, while the molar proportion of propionate decreased compared to the 67 BS and 100 BS groups with an increased proportion of barley in their diets (*p* < 0.001). The molar proportion of butyrate in the 33 BS group was higher than in the other groups (*p* < 0.001). The molar proportion of isobutyrate in the 33 BS group tended to be higher than that in 100 BS group (*p* = 0.073), while the molar proportion of isovalerate in the 100 BS group was lower than that in the CS, 33 BS and 67 BS groups (*p* < 0.001). The molar ratios of valerate in the CS and 33 BS groups were lower than that in the 67 BS and 100 BS groups (*p* = 0.035). The pH in the rumens of Hu sheep in the 100 BS group was lower relative to that of the CS and 67 BS groups (*p* = 0.026), but there was no difference in the ruminal pH between the 100 BS and 33 BS groups (*p* > 0.05.) There was no difference in the DMI among the groups (*p* = 0.386).

### 3.2. Diversity of Ruminal Bacterial Communities

Alpha diversity refers to the diversity in a specific environment or ecosystem, which is mainly used to reflect the richness and uniformity of species. The OTUs and Shannon indexes for the rumen microbial α-diversity of Hu sheep in the CS and 33 BS groups were higher than those of the 67 BS and 100 BS groups (*p* < 0.001), and the Chao1 and Ace indexes were higher than that of the 100 BS group (*p* < 0.003, Table 4). On the contrary, the Simpson indexes for the CS and 33 BS groups were lower than those of the 67 BS and 100 BS groups (*p* = 0.004). By increasing the ratio of barley in the diet, the OTUs and the Chao1, Ace and Shannon indexes all decreased linearly, but the Simpson index increased (*p* < 0.001). As the proportion of barley in the diet increased, the points that represented corresponding groups showed that the CS, 33 BS and 100 BS groups were far apart, and there were differences in species diversity (*p* < 0.05, Figure 1).

### 3.3. Abundances and Real-Time PCR Quantification of Bacteria

At the phylum level (with a relative abundance of at least one group > 0.1%), *Bacteroidetes*, *Firmicutes* and *Proteobacteria* were the dominant bacteria in the rumen, with the highest proportion of bacteria belonging to *Bacteroidetes*, followed by *Firmicutes* and *Proteobacteria*. In the present study, there were no differences in the proportions of *Actinobacteria* in the rumen between these groups (*p* > 0.05). When increasing the ratio of barley in the diet, the *Bacteroidetes* in the rumen showed a quadratic change (*p* = 0.004). The *Bacteroidetes* in the rumen of Hu sheep in the 33 BS group had the highest relative abundance, which was higher than that in the CS and 100 BS groups (*p* = 0.013, Table 5). The increase in barley in the diet reduced the relative abundance of *Saccharibacteria* in the rumen (*p* = 0.001). The relative abundance of *Saccharibacteria* in the CS group was not different from that of the 33 BS group, but it was higher than in the 67 BS and 100 BS groups (*p* = 0.009). The ratio of barley in the diet had no effect on the relative abundance of *Fibrobacteres* (*p* = 0.128); however, the numerical value of the relative abundance of *Fibrobacteres* in the rumen of the 33 BS Hu sheep was the highest. The relative abundance of *Firmicutes* in the CS group was higher than that in the other groups (*p* < 0.001), and the relative abundance of *Proteobacteria* was higher in the 67 BS and 100 BS groups than in the CS and 33 BS groups (*p* < 0.001). The relative abundance of *Spirochaetes* in the CS group had a higher trend than in the other groups (*p* < 0.055), and showed a linear decrease with an increase in the barley ratio in the diet (*p* = 0.017). Among the treatments, the relative abundance of *Synergistetes* showed a quadratic change (*p* = 0.046), and the relative abundance of *Synergistetes* in the 33 BS group was higher than in the other three groups (*p* = 0.006). The relative abundance of the acetate-producing *Verrucomicrobia* bacteria decreased linearly (*p* = 0.005) with an increase in barley, as seen by the greater relative abundance of these bacteria in the CS and 33 BS groups (*p* = 0.025).

At the genus level (with a relative abundance of at least one group > 0.1%), *Prevotella* was the dominant bacterial genus in the rumen of Hu sheep. The relative abundances of *Anaerovibrio* (*p* = 0.013) and *Bacteroides* (*p* = 0.037) in the CS and 33 BS groups were higher than those in the 100 BS group, and decreased linearly with an increase in the ratio of barley in the diet (*p* < 0.05, Table 6). The relative abundances of *Butyrivibrio* (*p* = 0.002) and *Desulfovibrio* (*p* = 0.005) in the CS group were higher than those in the 67 BS and 100 BS groups, and decreased linearly as the ratio of barley in the diet increased (*p* < 0.05). The relative abundances of *Mogibacterium*, *Moryella*, *Treponema* and *Stomatobaculum* in the CS group were higher than those in the other groups (*p* < 0.001), and the relative abundances of *Mogibacterium* and *Stomatobaculum* in the 33 BS group were higher than those in the 67 BS group and the 100 BS group (*p* < 0.001). *Prevotella* had the highest abundance in the rumen of Hu sheep in the 33 BS and 67 BS groups (*p* = 0.011), which increased as the ratio of barley in the diet increased (*p* = 0.002). *Pyramidobacter* was more abundant in the 100 BS group than in the other groups (*p* = 0.012). In addition, the relative abundances of *Ruminococcus* (*p* = 0.004) and *Saccharofermentans* (*p* = 0.049) in the 33 BS group were higher than those in the 67 BS group and 100 BS group, and increasing the ratio of barley in the diet reduced the relative abundances of these bacterial genera (*p* < 0.05). The content of *Fibrobacter succinogenes* in the rumen of the 33 BS group Hu sheep was higher than that of the CS group (*p* = 0.020, Table 7). In the results of this experiment, the content of *Ruminococcus flavefaciens* in the rumen of Hu sheep showed a quadratic change (*p* = 0.01), and the content was higher in the 33 BS group than in the other groups (*p* = 0.009). As the ratio of barley in the diet increased, the content of *Butyrivibrio fibrisolvens* in the rumen of Hu sheep showed a linear decrease (*p* = 0.069). The content of *Streptococcus bovis* in the rumen among these groups showed a quadratic change (*p* = 0.001), and the content of *Streptococcus bovis* in the 33 BS and 67 BS groups was higher than that in the CS group and 100 BS group (*p* = 0.009). Increasing the ratio of barley in the diet produced a linear increase in the content of *Selenomonas ruminantium*, *Prevotella brevis* and total bacteria in the ruminants (*p* < 0.05).

## 4. Discussion

Previous studies have shown that barley starch has a higher amylopectin content and is less bound with insoluble proteins, resulting in barley starch being more accessible to enzymatic hydrolysis compared with corn starch after the dry-rolled process (16.23%/h vs. 2.99%/h for the rate of degradation of the degradable fraction of starch) [17]. The ruminal pH decreases with an increasing proportion of barley starch in diets fed to cows. In the present study, the ruminal pH in the 100 BS group was lower than that of the other treatments, which was consistent with previous studies [12]. The rumen pH depends on the content of organic acids in the rumen. Barley starch is rapidly degraded by rumen microorganisms, which produces excessive organic acids, reducing the ruminal pH. The concentration of TVFA in the 67 BS group decreased compared to that of the 33 BS group, which was due to the fact that the DMI of the sheep in the 67 BS group was numerically the lowest, resulting in deficient organic acid accumulation. In this experiment, the molar proportions of acetate and the ratios of acetate to propionate in the 67 and 100 BS groups were lower than those in the other two groups, while the proportions of propionate and valerate increased. Previous studies have indicated that an increased level of dietary barley could increase the production of propionate and valerate, but decrease the molar ratio of acetate, due to higher content of RDS in barley compared to corn [9,18,19]. In this study, the molar proportion of butyrate in the 33 BS group was higher than that in the 67 BS and the 100 BS groups. *Pseudobutyrivibrio* and *Butyrivibrio fibrisolvens* are butyrate producers [20], but there were no superiorities for these bacteria in the 33 BS group compared to other groups. Zhang et al. showed that a high-concentrate diet increased rumen butyrate concentration, due to the fact that a high-concentrate diet increases the abundance of genes encoding pyruvate kinase, pyruvate ferredoxin oxidoreductase and L-lactate dehydrogenase [21]. These results were mainly attributed to the combination effect of 33% barley starch and 67% corn starch in the diet, which promoted the relative abundances of pyruvate kinase, pyruvate ferredoxin oxidoreductase and L-lactate dehydrogenase. However, there were no relevant results for the functional genes involved in the metabolic routes of VFAs in our study, and further study is needed.

The rapid fermentation of high-grain diets in the rumen can easily cause the accumulation of short-chain fatty acids and reduce the ruminal pH, which thereby changes the richness and diversity of microflora in the rumen [22,23]. The study conducted by Zhang et al. demonstrated that feeding a high-starch diet reduced both the richness and diversity of the bacterial community, with lower Ace, Chao 1 and Shannon indexes and a higher Simpson index [24]. In this study, the OTUs and the Chao1 and Ace indexes of the CS and 33 BS groups were higher than those of the 100 BS group. These indexes were positively correlated with the richness of rumen microflora, indicating that the rumen microbial richness of the 100 BS groups decreased. The decreased Shannon index and the increased Simpson index in the 67 BS group and 100 BS group indicated that the diversity of rumen microbes was decreased, which might be related to the lower pH in the rumen [13]. Compared with the starch in corn, the starch from wheat or barley contains a higher proportion of amylopectin and is usually coated with soluble proteins, which can be easily hydrolyzed by enzymes secreted by microorganisms [25,26]. Therefore, these changes in this study were considered to be associated with the increased barley starch level in the rumen-fermented substrates, which reduced the ruminal pH, especially in the 100 BS group (pH = 5.61). This was consistent with the results obtained by Plaizier et al. [27] and Hua et al. [28]. In addition, the composition of the rumen bacterial community among the four groups was also different, as indicated by the clear separation of sample points between the CS, 33 BS and 100 BS groups on the PCoA plots, which was in agreement with the findings of Zhang et al. [24]. These results indicated that replacing 33% of the corn with barley in the diet of Hu sheep can increase the richness and diversity of rumen microbes.

In the present study, *Bacteroidetes* were the most dominant bacteria at the phylum level in the rumen, which is consistent with previous studies in sheep [29]. The relative abundances of *Bacteroidetes* and *Bacteroides* in the 100 BS group decreased, which was attributed to the low ruminal pH, because many gram-negative *Bacteroidetes* are sensitive to pH [13,30]. Previous studies have shown that *Candidatus Saccharibacteria* and *Verrucomicrobia* can utilize the fiber fraction of the diet, but high-concentrate diet feeding would reduce the abundances of these bacteria due to an increased starch content [28,31]. Our results also showed that the relative abundances of *Candidatus Saccharibacteria*, *Verrucomicrobia* and *Synergistetes* decreased with an increasing barley starch level in the diet. In our study, the relative abundance of *Firmicutes* in the CS group was higher than other groups, which is due to the lower rumen starch degradation for corn than barley. There are greater degradable fractions of dry-rolled corn grain than dry-rolled barley grain that may have sustained ruminal fermentation over a longer duration, which promotes *Firmicutes* proliferation [30]. Previous studies have shown that the *Proteobacteria* are highly adaptable to dietary starch [32,33]. In our study, we also observed that the relative abundance of *Proteobacteria* improved with increasing dietary barley starch.

The relative abundances of *Bacteroides*, *Butyrivibrio*, *Pseudobutyrivibrio*, *Ruminococcus* and *Treponema* linearly decreased, which are sensitive to pH [34]. Changes for these bacteria would be due to the lower pH with an increasing proportion of dietary barley, especially in the 100 BS group. *R. albus* linearly increased with increasing dietary barley starch content. Although *R*. *albus* is a cellulolytic bacterium, the bacterium can utilize the intermediates (hexose and pentose) of starch breakdown [35]. Hence, the increase in *R*. *albus* in the rumen may have contributed to the sufficient rumen-degradable starch derived from the barley starch in the diet. However, once the rumen pH sharply decreased, the abundance of *Ruminococcus* declined [20], which also explained the changes in *Ruminococcus* in this study. The branched fatty acids (isobutyric, isovaleric, etc.) are the growth factors of cellulolytic bacteria in the rumen [36,37]. In the current study, the molar proportions of isobutyrate and isovalerate in the rumen in the 33 BS group were higher than those in the 100 BS group, which may promote the proliferation of *Fibrobacter succinogenes* and *Ruminococcus flavefaciens*. *Butyrivibrio fibrisolvens*, which is sensitive to rumen pH, is an important cellulolytic bacterium in the rumen. Thus, the decreased rumen pH of the 100 BS group would repress the proliferation of *Butyrivibrio fibrisolvens* [13]. The content of amylolytic bacteria, including *Prevotella*, *Streptococcus bovis*, *Selenomonas ruminantium* and *Prevotella brevis* showed a linear increase with increasing dietary BS levels in the present study. It may be that increasing the proportion of barley in the diet increased the content of rumen-degradable starch, which provided enough substrate for amylolytic bacteria, similar to the observations of Zhang et al. [21]. These results indicated that a high barley starch level in the diet promoted the proliferation of amylolytic bacteria, but the abundances of pH-sensitive bacteria decreased. Compared with the previous study by Ma et al. [8], more differences in the microflora of the rumen were observed in this trial. The reason for the differences between these studies could be due to the different animals, with fattening sheep of a similar age (about 3 months) and initial BW (29.70 ± 1.70 kg) used in the former, while fistula sheep of about 10 months age and an initial BW of 53.78 ± 3.14 kg were used in our study. Another reason could be a different feeding period (70 vs. 16 d).

## 5. Conclusions

Compared with a diet that obtains all of its starch from corn grain, the substitution of 33% corn starch with barley starch in the diet improved rumen fermentation, but did not affect the rumen fermentation pattern. In addition, the 33 BS group experienced an increase in the richness and diversity of rumen microflora compared with the 67 BS and 100 BS groups, as well as an increase in the relative abundances of bacteria belonging to the phylum *Bacteroidetes* and genus *Prevotella*, the content of cellulolytic bacteria (*Fibrobacter succinogenes* and *Ruminococcus flavefaciens*) and the content of amylolytic bacteria (*Streptococcus bovis*, *Selenomonas ruminantium* and *Prevotella brevis*) relative to the CS group. Therefore, it is suggested to replace 33% of the corn with barley in the diet for Hu sheep.

## Figures and Tables

**Figure 1 animals-12-01941-f001:**
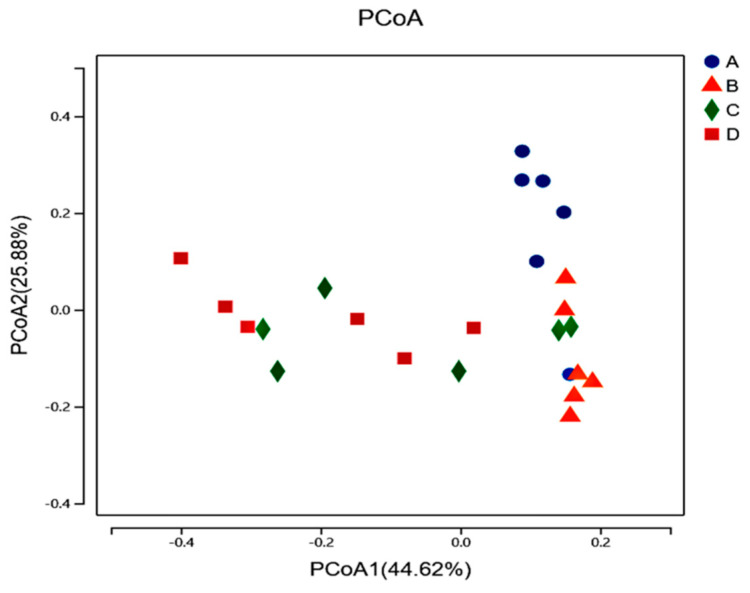
Analysis of β−diversity at 2 h after morning feeding among different treatment groups. “A” represents the CS group, with 100% of starch derived from corn; “B” represents the 33 BS group, with 33% of starch derived from barley + 67% of starch derived from corn; “C” represents the 67 BS group, with 67% of starch derived from barley + 33% of starch derived from corn; “D” represents the 100 BS group, with 100% of starch derived from barley.

**Table 1 animals-12-01941-t001:** Ingredients and chemical composition of diets.

Items	Barley ^1^			
	CS	33 BS	67 BS	100 BS
Ingredients (% air-dried form)				
Barley straw	15.00	15.00	15.00	15.00
Corn	37.00	22.00	12.00	0.00
Barley	0.00	16.20	32.00	48.00
Corn bran	24.00	17.50	16.80	15.50
Cottonseed meal	7.50	6.00	6.60	6.00
Soybean meal	9.00	7.00	8.00	8.00
Corn gluten feed	0.00	8.80	2.10	0.00
Molasses	4.00	4.00	4.00	4.00
Sodium bicarbonate	0.60	0.60	0.60	0.60
Limestone	1.20	1.20	1.20	1.20
Salt	0.70	0.70	0.70	0.70
Premix ^2^	1.00	1.00	1.00	1.00
Total	100.00	100.00	100.00	100.00
Nutrient content (% DM)				
DM	91.48	91.40	91.32	91.81
CP	13.72	13.57	13.81	13.69
NDF	30.59	30.96	30.75	31.25
ADF	14.94	14.48	14.35	14.28
Starch	23.99	23.90	24.15	24.02
Ca	0.87	0.85	0.84	0.83
P	0.26	0.30	0.30	0.31
Metabolizable energy (MJ/kg)	9.81	9.84	9.69	9.57

^1^ Dietary barley levels defined by the proportion of barley starch in diets: CS, 100% of starch derived from corn; 33 BS, 33% of starch derived from barley + 67% of starch derived from corn; 67 BS, 67% of starch derived from barley + 33% of starch derived from corn; 100 BS, 100% of starch derived from barley. ^2^ The premix provided the following per kg of diet: Fe, 25 mg; Mn, 40 mg; Zn, 40 mg; Cu, 8 mg; I, 0.3 mg; Se, 0.2 mg; Co, 0.1 mg; VA, 940 IU; VD, 111 IU; VE, 20 IU.

**Table 2 animals-12-01941-t002:** Rumen microorganism quantitative real-time PCR amplification primers.

Species	Primer Sequence (5′-3′) ^1^
*Fibrobacter succinogenes*	F ^2^: GGTATGGGATGAGCTTGC
	R ^3^: GCCTGCCCCTGAACTATC
*Ruminococcus albus*	F: TGTTAACAGAGGGAAGCAAAGCA
	R: TGCAGCCTACAATCCGAACTAA
*Ruminococcus flavefaciens*	F: CGAACGGAGATAATTTGAGTTTACTTAGG
	R: CGGTCTCTGTATGTTATGAGGTATTACC
*Butyrivibrio fibrisolvens*	F: GCCTCAGCGTCAGTAATCG
	R: GGAGCGTAGGCGGTTTTAC
*Streptococcus bovis*	F: TTCCTAGAGATAGGAAGTTTCTTCGG
	R: ATGATGGCAACTAACAATAGGGGT
*Prevotella brevis*	F: GGTTCTGAGAGGAAGGTCCCC
	R: TCCTGCACGCTACTTGGCTG
*Selenomonas ruminantium*	F: CAATAAGCATTCCGCCTGGG
	R: TTCACTCAATGTCAAGCCCTGG

^1^ The primer sequences are referenced from Zhang et al. [16]; ^2^ F means forward primers; ^3^ R means reverse primers.

**Table 3 animals-12-01941-t003:** Effects of different barley ratios in diets on volatile fatty acids in the rumen of Hu sheep.

Items	Time/h	Treatment				SEM	*p*-Value		
		CS	33 BS	67 BS	100 BS		Treatment	Time	Interaction
TVFA, mmol/L	0	83.68	91.00	55.43	71.31	2.122	0.007	0.071	0.599
	2	89.91	100.86	107.16	89.44				
	4	72.22	97.93	77.41	92.45				
	6	90.93	109.24	81.79	88.90				
	8	86.14	99.60	66.77	97.29				
	10	96.28	101.65	88.85	80.34				
Acetate, %	0	66.11	66.35	66.79	65.62	0.498	<0.001	0.007	0.692
	2	64.52	65.26	58.41	53.05				
	4	61.47	64.04	60.10	55.01				
	6	63.62	64.39	59.17	61.18				
	8	63.59	63.86	58.11	59.02				
	10	63.72	64.36	59.45	61.21				
Propionate, %	0	20.84	20.08	22.12	22.56	0.620	<0.001	0.159	0.877
	2	21.37	18.75	29.81	33.27				
	4	23.12	20.68	28.25	34.53				
	6	23.24	20.43	28.68	26.96				
	8	23.13	21.12	30.62	30.02				
	10	22.77	20.18	28.52	26.99				
Isobutyrate, %	0	0.74	0.88	1.15	1.36	0.036	0.073	<0.001	0.631
	2	0.90	0.51	0.67	1.16				
	4	0.59	0.43	0.58	0.41				
	6	0.56	0.30	0.77	0.57				
	8	0.54	0.39	0.41	0.38				
	10	0.61	0.35	0.49	0.57				
Butyrate, %	0	10.03	10.69	7.48	7.88	0.247	<0.001	0.200	0.997
	2	10.78	13.78	8.93	10.12				
	4	12.56	13.26	9.12	8.29				
	6	10.61	13.48	9.10	9.00				
	8	10.71	13.16	8.93	8.82				
	10	10.88	13.50	9.65	9.11				
Isovalerate, %	0	1.39	1.11	1.55	1.55	0.036	<0.001	<0.001	0.834
	2	1.41	0.77	0.97	1.07				
	4	1.07	0.58	0.79	0.64				
	6	1.00	0.46	1.04	1.08				
	8	1.01	0.56	0.62	0.71				
	10	1.05	0.57	0.80	1.01				
Valerate, %	0	0.89	0.88	0.92	1.03	0.027	0.035	0.429	0.975
	2	1.02	0.93	1.22	1.33				
	4	1.20	1.00	1.15	1.12				
	6	0.98	0.94	1.23	1.20				
	8	1.01	0.91	1.30	1.05				
	10	0.97	1.04	1.09	1.11				
Acetate:Propionate	0	3.55	3.79	3.24	3.04	0.079	<0.001	0.110	0.996
	2	3.32	3.75	2.27	2.00				
	4	2.97	3.27	2.57	1.83				
	6	2.91	3.38	2.56	2.55				
	8	2.98	3.20	2.26	2.29				
	10	2.92	3.40	2.39	2.42				
pH	0	6.12	6.24	6.56	6.43	0.032	0.026	<0.001	0.418
	2	5.83	5.79	5.79	5.50				
	4	5.92	5.61	5.54	5.37				
	6	5.62	5.45	5.72	5.45				
	8	5.85	5.75	5.83	5.49				
	10	5.70	5.66	5.85	5.63				
DMI, kg/d	-	1.14	1.41	0.87	1.06	0.112	0.386	-	-

Dietary barley levels defined by the proportion of barley starch in diets: CS, 100% of starch derived from corn; 33 BS, 33% of starch derived from barley + 67% of starch derived from corn; 67 BS, 67% of starch derived from barley + 33% of starch derived from corn; 100 BS, 100% of starch derived from barley.

**Table 4 animals-12-01941-t004:** Analysis of α-diversity at 2 h after morning feeding among different treatment groups.

Items	Treatments ^1^				SEM	*p*-Value		
	CS	33 BS	67 BS	100 BS		Treatment	Linear	Quadratic
OTUs	740.83 ^a^	699.00 ^a^	555.50 ^b^	475.50 ^b^	31.550	0.003	<0.001	0.694
Chao1	872.94 ^a^	845.61 ^ab^	704.07 ^bc^	585.94 ^c^	34.831	0.004	<0.001	0.411
Ace	861.46 ^a^	845.40 ^a^	703.74 ^ab^	581.49 ^b^	34.098	0.003	<0.001	0.323
Shannon	4.57 ^a^	4.13 ^a^	3.43 ^b^	3.38 ^b^	0.137	<0.001	<0.001	0.314
Simpson	0.03 ^b^	0.06 ^b^	0.12 ^a^	0.11 ^a^	0.004	0.004	0.001	0.340
Coverage	0.996	0.996	0.997	0.997	0.000	0.063	0.015	0.246

^1^ Dietary barley levels defined by the proportion of barley starch in diets: CS, 100% of starch derived from corn; 33 BS, 33% of starch derived from barley + 67% of starch derived from corn; 67 BS, 67% of starch derived from barley + 33% of starch derived from corn; 100 BS, 100% of starch derived from barley. ^a–c^ Values within a row with uncommon letters differ significantly among the four groups (*p* < 0.05).

**Table 5 animals-12-01941-t005:** Effects of dietary barley starch levels on relative abundances of dominant phyla at 2 h after morning feeding in the rumen of Hu sheep.

Items	Treatments ^1^				SEM	*p*-Value		
	CS	33 BS	67 BS	100 BS		Treatment	Linear	Quadratic
*Actinobacteria*	0.04	0.03	0.02	0.03	0.005	0.591	0.264	0.459
*Bacteroidetes*	53.59 ^b^	71.44 ^a^	61.48 ^ab^	51.44 ^b^	2.565	0.013	0.397	0.004
*Saccharibacteria*	0.49 ^a^	0.35 ^ab^	0.14 ^bc^	0.03 ^c^	0.057	0.009	0.001	0.854
*Fibrobacteres*	0.40	1.64	0.69	0.61	0.210	0.128	0.966	0.092
*Firmicutes*	29.18 ^a^	16.36 ^b^	10.78 ^b^	10.24 ^b^	1.843	0.000	0.000	0.009
*Proteobacteria*	3.01 ^b^	3.44 ^b^	28.30 ^a^	35.33 ^a^	4.092	0.000	0.000	0.533
*Spirochaetes*	1.45	0.70	0.63	0.54	0.141	0.055	0.017	0.209
*Synergistetes*	0.61 ^b^	3.27 ^a^	0.15 ^b^	0.08 ^b^	0.406	0.006	0.119	0.046
Unclassified	3.48 ^a^	1.47 ^b^	0.71 ^bc^	0.29 ^c^	0.311	0.000	0.000	0.032
*Verrucomicrobia*	1.58 ^a^	1.58 ^a^	0.47 ^b^	0.07 ^c^	0.231	0.025	0.005	0.615

^1^ Dietary barley levels defined by the proportion of barley starch in diets: CS, 100% of starch derived from corn; 33 BS, 33% of starch derived from barley + 67% of starch derived from corn; 67 BS, 67% of starch derived from barley + 33% of starch derived from corn; 100 BS, 100% of starch derived from barley. ^a–c^ Values within a row with uncommon letters differ significantly among the four groups (*p* < 0.05).

**Table 6 animals-12-01941-t006:** Effects of dietary barley starch levels on relative abundances of dominant genera at 2 h after morning feeding in the rumen of Hu sheep.

Items	Treatments ^1^				SEM	*p*-Value		
	CS	33 BS	67 BS	100 BS		Treatment	Linear	Quadratic
*Anaerovibrio*	0.46 ^a^	0.39 ^a^	0.27 ^ab^	0.02 ^b^	0.055	0.013	0.002	0.298
*Bacteroides*	0.38 ^a^	0.42 ^a^	0.30 ^ab^	0.15 ^b^	0.038	0.037	0.011	0.165
*Barnesiella*	6.51	2.34	0.29	2.22	0.889	0.067	0.042	0.073
*Butyrivibrio*	1.09 ^a^	0.70 ^ab^	0.27 ^b^	0.29 ^b^	0.097	0.002	0.000	0.160
*Centipeda*	0.68	1.46	0.08	0.16	0.228	0.092	0.122	0.282
*Clostridium_XlVa*	0.32	0.11	0.34	0.046	0.052	0.090	0.171	0.664
*Desulfovibrio*	0.24 ^a^	0.15 ^ab^	0.06 ^bc^	0.03 ^c^	0.025	0.005	0.001	0.487
*Eubacterium*	0.21	0.19	0.13	0.03	0.031	0.136	0.026	0.502
*Fibrobacter*	0.40	1.64	0.69	0.61	0.210	0.128	0.966	0.092
*Lachnospirace_incertae_sedis*	0.17	0.19	0.07	0.13	0.029	0.535	0.395	0.701
*Mogibacterium*	0.07 ^a^	0.02 ^b^	0.01^c^	0.01^c^	0.006	0.000	0.000	0.006
*Moryella*	0.29 ^a^	0.07 ^b^	0.02 ^b^	0.07 ^b^	0.034	0.019	0.017	0.027
*Prevotella*	31.39 ^b^	55.82 ^a^	46.07 ^a^	44.44 ^ab^	2.795	0.011	0.165	0.010
*Pseudobutyrivibrio*	0.46	0.15	0.21	0.01	0.062	0.069	0.019	0.619
*Pyramidobacter*	0.01 ^b^	0.02 ^b^	0.04 ^b^	0.08 ^a^	0.009	0.012	0.002	0.218
*Ruminococcus*	0.34 ^a^	0.33 ^a^	0.13 ^b^	0.04 ^b^	0.038	0.004	0.001	0.519
*Saccharibacteria*	0.49 ^a^	0.35 ^ab^	0.14 ^bc^	0.03 ^c^	0.057	0.009	0.001	0.854
*Saccharofermentans*	0.33 ^ab^	0.35 ^a^	0.14 ^b^	0.14 ^b^	0.037	0.049	0.015	0.972
*Selenomonas*	0.57	0.73	2.07	0.83	0.241	0.093	0.297	0.127
*Sphaerochaeta*	0.03 ^b^	0.06 ^b^	0.56 ^a^	0.07 ^b^	0.073	0.020	0.283	0.049
*Stomatobaculum*	0.30 ^a^	0.13 ^b^	0.02^c^	0.03^c^	0.028	0.000	0.000	0.020
*Succiniclasticum*	1.14	0.62	0.55	0.87	0.120	0.287	0.409	0.083
*Succinivibrio*	1.14	2.40	0.80	1.07	0.310	0.230	0.393	0.410
*Treponema*	1.41 ^a^	0.63 ^b^	0.60 ^b^	0.46 ^b^	0.133	0.037	0.012	0.180
Unclassified	42.20	24.60	39.49	41.28	2.917	0.101	0.619	0.086

^1^ Dietary barley levels defined by the proportion of barley starch in diets: CS, 100% of starch derived from corn; 33 BS, 33% of starch derived from barley + 67% of starch derived from corn; 67 BS, 67% of starch derived from barley + 33% of starch derived from corn; 100 BS, 100% of starch derived from barley. ^a–c^ Values within a row with uncommon letters differ significantly among the four groups (*p* < 0.05).

**Table 7 animals-12-01941-t007:** Effects of dietary barley starch levels on the rumen microbial content at 2 h after morning feeding of Hu sheep (Log10 16S rRNA copy number/mL rumen fluid).

Items	Treatments ^1^				SEM	*p*-Value		
	CS	33 BS	67 BS	100 BS		Treatment	Linear	Quadratic
*Ruminococcus albus*	8.22	8.21	8.75	9.22	0.17	0.100	0.020	0.457
*Fibrobacter succinogenes*	7.01 ^b^	8.58 ^a^	7.72 ^ab^	8.00 ^ab^	0.18	0.020	0.122	0.044
*Ruminococcus flavefaciens*	8.93 ^b^	10.30 ^a^	9.30 ^b^	9.15 ^b^	0.17	0.009	0.789	0.011
*Butyrivibrio fibrisolvens*	9.20	9.14	9.03	8.32	0.17	0.215	0.069	0.323
*Streptococcus bovis*	5.47 ^b^	6.11 ^a^	5.90 ^a^	5.44 ^b^	0.09	0.009	0.663	0.001
*Selenomonas ruminantium*	8.74 ^b^	9.37 ^a^	9.08 ^ab^	9.37 ^a^	0.09	0.022	0.028	0.270
*Prevotella brevis*	11.70 ^b^	12.25 ^a^	12.15 ^a^	12.37 ^a^	0.07	0.002	0.001	0.148
Total bacteria	13.34 ^b^	14.51 ^a^	14.12 ^ab^	14.84 ^a^	0.19	0.026	0.011	0.508

^1^ Dietary barley levels defined by the proportion of barley starch in diets: CS, 100% of starch derived from corn; 33 BS, 33% of starch derived from barley + 67% of starch derived from corn; 67 BS, 67% of starch derived from barley + 33% of starch derived from corn; 100 BS, 100% of starch derived from barley. ^a,b^ Values within a row with uncommon letters differ significantly among the four groups (*p* < 0.05).

## Data Availability

The data presented in this study are available on request from the corresponding author.

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
