# Peer review of "Effects of Barley Starch Level in Diet on Fermentation and Microflora in Rumen of Hu Sheep"

_animals, 2022, doi:10.3390/ani12151941_

Round 1

Reviewer 1 Report

General comment

The manuscript entitled “Effects of barley starch level in diet on fermentation and microflora in rumen of Hu sheep” by Zhian Zhang et al. investigated the effects of rapidly fermentable barley starch on the rumen fermentation and microbiota. The authors found that barley starch did not affect the rumen fermentation pattern and increase the diversity of rumen microbiota. This study provides a feasible strategy for improving the production performance of ruminants. On the other hand, I think that the manuscript still has some unclear descriptions and problems as indicated below. Therefore, the present manuscript needs some revisions prior to its publication.

Major comments

l  All results: This study investigated the rumen fermentation and microbiota at 0, 2, 4, 6, 8, 10, and 12 h, however all results show only one value. Do the results represent the mean or the representative value? I think that the temporal dynamics of rumen fermentation and microbiota are important to evaluate the rapidly fermentable barley starch. I recommend that the results at all sampling points should be listed for improving the importance of your manuscript. 

l  Line 106: The manuscript describes that 32 sheep used in this study were fed twice at 0800 and 1800 h, and rumen fluid was collected at 0, 2, 4, 6, 8, 10, and 12 h after morning feeding. I am worried the results at 10 and 12 h after morning were influenced by evening feeding.

l  Line 138: Information about the primer, such as primer name (i.e., 341F and 806R), primer sequence, and reference, should be described in the Materials and Methods.

l  Line 146: The processing condition of sequence data should be also described (i.e., threshold of quality score, similarity for generating OTUs, and database for taxonomic classification).

l  L156: Why did you use the species-level relative abundance for calculating the diversity? I think that the most OTUs cannot be classified into the species-level. Additionally, the description of beta-diversity (PCoA) should be added (i.e., Bray-Curtis or Jaccard).

l  Line 208: Please add the discussion why did the Simpson index increased in the 67BS and 100BS.

l  Results and Discussion: The authors describe both the relative abundance by MiSeq and the absolute abundance by qPCR as just “the abundance”. The results of the relative abundance by MiSeq and the results of the absolute abundance by qPCR should be clearly distinguished because these abundances are not completely linked with each other.

l  Data Availability: Sequencing data must be deposited to the public database unless there are some particular reasons.

Minor comments

l  Table 1 : TVFA, mmol => TVFA, mmol/L

l  Figure 1: Analysis of α-diversity => Analysis of β-diversity

l  Please confirm all abbreviations because there are some errors as the follows;

Line 152: OUT => OTU

Line 362: Please add the full form of RDS.

Author Response

Dear reviewer,

Thanks for your suggestions on manuscript (ID: animals-1759572) entitled “Effects of barley starch level in diet on fermentation and microflora in rumen of Hu sheep”. The authors have modified the manuscript as the suggestions and comments. The modified words were highlighted in red color in the revised manuscript. We deeply appreciate your suggestions and comments that help us improve the quality of the manuscript. The following is our response to the reviewer's comments.

Response to Reviewer 1 Comments

Point 1: All results: This study investigated the rumen fermentation and microbiota at 0, 2, 4, 6, 8, 10, and 12 h, however all results show only one value. Do the results represent the mean or the representative value? I think that the temporal dynamics of rumen fermentation and microbiota are important to evaluate the rapidly fermentable barley starch. I recommend that the results at all sampling points should be listed for improving the importance of your manuscript.

Response 1: Thanks for your suggestion. The results at all sampling points should be listed in this study (Table 3).

Point 2: Line 106: The manuscript describes that 32 sheep used in this study were fed twice at 0800 and 1800 h, and rumen fluid was collected at 0, 2, 4, 6, 8, 10, and 12 h after morning feeding. I am worried the results at 10 and 12 h after morning were influenced by evening feeding.

Response 2: Thanks for your suggestion. In this study, the second feeding time was 10 hours after the morning feeding, but after the rumen fluid was collected at this time point. In order to avoid the effect of evening feeding on rumen fermentation, we removed the corresponding data for the 12th hour after morning feeding.

Point 3: Line 138: Information about the primer, such as primer name (i.e., 341F and 806R), primer sequence, and reference, should be described in the Materials and Methods.

Response 3: Thanks for your suggestion. Information about primers has been supplemented in Materials and methods, and the modified words were highlighted in red color in the revised manuscript.

Point 4: Line 146: The processing condition of sequence data should be also described (i.e., threshold of quality score, similarity for generating OTUs, and database for taxonomic classification).

Response 4: Thanks for your suggestion. The processing condition of sequence data has been supplemented in Materials and methods (2.4. Extraction of bacterial DNA and Illumina MiSeq sequencing), and the modified words were highlighted in red color in the revised manuscript.

Point 5: L156: Why did you use the species-level relative abundance for calculating the diversity? I think that the most OTUs cannot be classified into the species-level. Additionally, the description of beta-diversity (PCoA) should be added (i.e., Bray-Curtis or Jaccard).

Response 5: I totally agree with you. The incorrect sentence has been corrected, and the modified words were highlighted in red color in the revised manuscript. To compare the dissimilarities in bacterial communities among the four groups, the principal coordinate analysis (PCoA) based on weighted UniFrac distances at OTU level was performed.

Point 6: Line 208: Please add the discussion why did the Simpson index increased in the 67BS and 100BS.

Response 6: Thanks for your suggestion. The reason why the Simpson index increased in the 67 BS and 100 BS has been added in the discussion. Compared with the starch in corn, the starch from wheat or barley contains a higher proportion of amylopectin and usually coats with soluble protein, which can be easily hydrolyzed by enzymes secreted by microorganisms. Therefore, these changes in this study were considered to be associated with the increased barley starch level in the rumen fermented substrates, which reduced the ruminal pH, especially in the 100 BS group (pH=5.61), which was consistent with the results obtained by Plaizier et al.

Point 7: Results and Discussion: The authors describe both the relative abundance by MiSeq and the absolute abundance by qPCR as just “the abundance”. The results of the relative abundance by MiSeq and the results of the absolute abundance by qPCR should be clearly distinguished because these abundances are not completely linked with each other.

Response 7: Thanks for your suggestion. The relative abundance by MiSeq and the absolute abundance have been distinguished in the results and discussion, and the modified words were highlighted in red color in the revised manuscript.

Point 8: Data Availability: Sequencing data must be deposited to the public database unless there are some particular reasons.

Response 8: Thanks for your suggestion. All raw sequences were deposited in the NCBI Sequence Read Archive (SRA) database and can be accessed via accession number: PRJNA859264.

Point 9: Table 1 : TVFA, mmol => TVFA, mmol/L;

Response 9: Thanks for your suggestion. TVFA, mmol has been corrected to TVFA, mmol/L, and the modified words were highlighted in red color in the revised manuscript.

Point 10: Figure 1: Analysis of α-diversity => Analysis of β-diversity

Response 10: Thanks for your suggestion. Analysis of α-diversity has been corrected to Analysis of β-diversity.

Point 11: Line 152: OUT => OTU

Response 11: Thanks for your suggestion. OUT has been corrected to OUT.

Point 12: Line 362: Please add the full form of RDS.

Response 12: Thanks for your suggestion. The full form of RDS has been added, and the modified words were highlighted in red color in the revised manuscript.

Reviewer 2 Report

I am partly satisfied with the corrections of the article. However, minimal correction is necessary:

Line 182 and 192 – group names differ. In the text „66 BS“, in the Table „67 BS“. Group names must be uniform in whole article.

Line 244 – replace “BS” with “33 BS”.

Author Response

Dear reviewer,

Thanks for your suggestions on manuscript (ID: animals-1759572) entitled “Effects of barley starch level in diet on fermentation and microflora in rumen of Hu sheep”. The authors have modified the manuscript as the suggestions and comments. The modified words were highlighted in red color in the revised manuscript. We deeply appreciate your suggestions and comments that help us improve the quality of the manuscript. The following is our response to the reviewer's comments.

Response to Reviewer 2 Comments

Point 1: Line 182 and 192 – group names differ. In the text „66 BS“, in the Table „67 BS“. Group names must be uniform in whole article.

Response 1: Thanks for your suggestion. 66 BS has been corrected to 67 BS, and the modified words were highlighted in red color in the revised manuscript.

Point 2: Line 244 – replace “BS” with “33 BS”

Response 2: Thanks for your suggestion. “BS” in Line 244 has been replaced by “33 BS”.

Round 2

Reviewer 1 Report

The authors have responded to all the comments, and the manuscript has been revised well. I think this manuscript will be acceptable after the following correction have been done: the results of microbial analysis at all sampling points are not needed to list, however the authors should describe which timepoint the microbiota represents in the Materials and methods or captions (for example, Fig 1. Analysis of β-diversity at 4 h after feeding; Table 5. relative abundances of dominant genera at 8 h after feeding; Table 6 caption. Values are shown as the mean relative abundances at all sampling points).

Author Response

Dear reviewer,

Thanks for your suggestions on manuscript (ID: animals-1759572) entitled “Effects of barley starch level in diet on fermentation and microflora in rumen of Hu sheep”. The authors have modified the manuscript as the suggestions and comments. The modified words were highlighted in red color in the revised manuscript. We deeply appreciate your suggestions and comments that help us improve the quality of the manuscript. The following is our response to the reviewer's comments.

Response to Reviewer 1 Comments

Point 1: The authors have responded to all the comments, and the manuscript has been revised well. I think this manuscript will be acceptable after the following correction have been done: the results of microbial analysis at all sampling points are not needed to list, however the authors should describe which timepoint the microbiota represents in the Materials and methods or captions (for example, Fig 1. Analysis of β-diversity at 4 h after feeding; Table 5. relative abundances of dominant genera at 8 h after feeding; Table 6 caption. Values are shown as the mean relative abundances at all sampling points).

Response 1: Thanks for your suggestion. The timepoint (2 h after morning feeding) the microbiota represents have been supplemented in the Materials and methods or captions, and the modified words were highlighted in red color in the revised manuscript.

This manuscript is a resubmission of an earlier submission. The following is a list of the peer review reports and author responses from that submission.

Round 1

Reviewer 1 Report

The paper is of scientific interest and falls within the scope of the journal. In general the experimental design and the analytical methods appropriate and the results obtained consistent.

Going to the previous paper of the authors (Ma, X., Zhou, W., Guo, T., Li, F., Li, F., Ran, T., ... & Guo, L. (2021). Effects of dietary barley starch contents on the performance, nutrient digestion, rumen fermentation and bacterial community of fattening Hu sheep. Frontiers in Nutrition, 1302) it is not clear to me if the objective of substituting corn by barley starch is in this second paper is based on economical or performance parameters. In both (the published and this ms) the conclusion in the abstract and the general conclusions do not fit very well and should be refreshed to give the reader a clearer message.

Author Response

Dear reviewer,

Thanks for your suggestions on manuscript (ID: animals-1681283) entitled “Effects of barley starch level in diet on fermentation and microflora in rumen of Hu sheep”. The authors have modified the manuscript as the suggestions and comments. The modified words were highlighted in red color in the revised manuscript. We deeply appreciate your suggestions and comments that help us improve the quality of the manuscript. The following is our response to the reviewer's comments.

Response to Reviewer 1 Comments

Point 1: “The paper is of scientific interest and falls within the scope of the journal. In general the experimental design and the analytical methods appropriate and the results obtained consistent. Going to the previous paper of the authors (Ma, X., Zhou, W., Guo, T., Li, F., Li, F., Ran, T., ... & Guo, L. (2021). Effects of dietary barley starch contents on the performance, nutrient digestion, rumen fermentation and bacterial community of fattening Hu sheep. Frontiers in Nutrition, 1302) it is not clear to me if the objective of substituting corn by barley starch is in this second paper is based on economical or performance parameters. In both (the published and this ms) the conclusion in the abstract and the general conclusions do not fit very well and should be refreshed to give the reader a clearer message.”

Response 1: Thanks for your suggestion. Our previous study has shown that the effects of replacing CS with BS in the diet on performance, nutrient digestion and rumen fermentation at a time point, one main reason for which might be due to the changes of rumen microflora. However, insufficient attention has been paid to the effects on rumen microflora in the previous study, a few bacteria were determined, which could not reflect alteration of rumen microbial structure. Further study is needed to clarify the effects of barley starch replacing corn starch on rumen microbial diversity, richness and bacteria that dominate at the phylum or genus level.

Point 2: The conclusion in the abstract do not fit very well and should be refreshed to give the reader a clearer message.

Response 2: Thanks for your suggestion. The conclusion has been refreshed, and the modified words were highlighted in red color in the revised manuscript.

Reviewer 2 Report

Dear authors,

I appreciate the work you must do for gaining these interesting results. I have only few comments.

Replace all tables that they are not on two sites. And the results must be described more accurately with an emphasis placed on the significant differences between groups.

Line 97 – uniform VA, VD and VE in the footnotes of table 1.

Line 174-175 – According to Table 3. the rumen pH value in 100 BS group is similar to 33 BS group. I think that sentence: “Compared to other groups, the pH in the rumen of Hu 174 sheep in the 100 BS group was lower” is incorrect.

Line 228-230 – From Table 5 is clear significant difference in Verrucomicrobia (groups CS and 33 BS) compared to (group 67 BS). However, the text of manuscript claims no significant difference. Eighter information in the Table 5, or information in the tex must be wrong.

Line 338 - [30] not with superior index.

Author Response

Dear reviewer,

Thanks for your suggestions on manuscript (ID: animals-1681283) entitled “Effects of barley starch level in diet on fermentation and microflora in rumen of Hu sheep”. The authors have modified the manuscript as the suggestions and comments. The modified words were highlighted in red color in the revised manuscript. We deeply appreciate your suggestions and comments that help us improve the quality of the manuscript. The following is our response to the reviewer's comments.

Response to Reviewer 2 Comments

Point 1: “Line 97 – uniform VA, VD and VE in the footnotes of table 1”

Response 1: Thanks for your suggestion. VA, VD and VE in the footnotes of table 1 have been uniformed.

Point 2: Line 174-175 – According to Table 3. the rumen pH value in 100 BS group is similar to 33 BS group. I think that sentence: “Compared to other groups, the pH in the rumen of Hu sheep in the 100 BS group was lower” is incorrect.

Response 2: Thanks for your suggestion. The incorrect sentence: “Compared to other groups, the pH in the rumen of Hu sheep in the 100 BS group was lower” has been corrected, and the modified words were highlighted in red color in the revised manuscript.

Point 3: Line 228-230 – From Table 5 is clear significant difference in Verrucomicrobia (groups CS and 33 BS) compared to (group 67 BS). However, the text of manuscript claims no significant difference. Eighter information in the Table 5, or information in the tex must be wrong.

Response 3: Thanks for your suggestion. The incorrect expression has been corrected, the sentence is shown like that “Increasing the ratio of barley in the diet, the abundance of acetate-producing bacteria Verrucomicrobia decreased linearly (p = 0.005), the abundance of Verrucomicrobia in the CS and 33 BS groups was higher than that in the 67 BS and 100 BS groups (p = 0.025)”, which is highlighted in red color in the revised manuscript.

Point 4: Line 338 - [30] not with superior index.

Response 4: Thanks for your suggestion. Reference [30] has been replaced by new one.

Reviewer 3 Report

Effect of replacing corn starch with barley starch, higher degradability in the rumen, in Hu sheep diet on ruminal fermentation end-products and bacteria community was conducted. Although this manuscript is well in presentation of the result especially numeration, identification & classification, this work very close to previous study (self-plagiarism & -citation). Those Ma et al. (Front. Nutr. 8:797801, 2022) and the present work did similar in treatments, Hu sheep, and detected parameter (ruminal fermentation end-products & bacteria abundance).

Author Response

Dear reviewer,

Thanks for your suggestions on manuscript (ID: animals-1681283) entitled “Effects of barley starch level in diet on fermentation and microflora in rumen of Hu sheep”. The authors have modified the manuscript as the suggestions and comments. The modified words were highlighted in red color in the revised manuscript. We deeply appreciate your suggestions and comments that help us improve the quality of the manuscript. The following is our response to the reviewer's comments.

Response to Reviewer 3 Comments

Point 1: Effect of replacing corn starch with barley starch, higher degradability in the rumen, in Hu sheep diet on ruminal fermentation end-products and bacteria community was conducted. Although this manuscript is well in presentation of the result especially numeration, identification & classification, this work very close to previous study. Those Ma et al. (Front. Nutr. 8:797801, 2022) and the present work did similar in treatments, Hu sheep, and detected parameter (ruminal fermentation end-products & bacteria abundance).

Response 1: Thanks for your suggestion. Our previous study (Ma et al., 2022) has shown that the effects of replacing CS with BS in the diet on performance, nutrient digestion and rumen fermentation at a time point, one main reason for which might be due to the changes of rumen microflora. However, insufficient attention has been paid to the effects on rumen microflora in the previous study, a few bacteria were determined, which could not reflect alteration of rumen microbial structure. Further study is needed to clarify the effects of barley starch replacing corn starch on rumen microbial diversity, richness and bacteria that dominate at the phylum or genus level.

Point 2: The methods and results in the study should be clearly presented.

Response 2: Thanks for your suggestion. Relevant contents have been modified in the manuscript, and the modified words were highlighted in red color in the revised manuscript.

What we would like to add here is that compared with the previous study (Ma et al., 2022), the animals used are also different. The former used fattening sheep, while in our study,  sheep equipped with fistula  were used. The dynamic pH was measured in fistula sheep in the study of Ma et al.(2022), and some differences between different groups were also found. On this basis, we further conducted a short-term trial to explore the changes of rumen fermentation and microflora of sheep, which was not only a supplement to the previous experiments, but also a further exploration of the reasons for the differences in dynamic pH by analyzing the effects of barley starch replacing corn starch on rumen microbial diversity, richness and bacteria that dominate at the phylum or genus level.

Round 2

Reviewer 3 Report

Self-plagiarism